# Effects of human development, primary care coverage and the COVID-19 pandemic on the fertility rate among adolescents in Brazil in the period 2012–2021

**Rosa Maria Soares Madeira Domingues**[1☯*], **Rafael Sant´Ana Herzog**[2☯],
**Marcos Augusto Bastos Dias**[3], **Rossana Pulcineli Vieira Francisco**[4],
**Agatha Sacramento Rodrigues**[2☯]

**1** Instituto Nacional de Infectologia Evandro Chagas, Rio de Janeiro (RJ), Brasil, **2** Universidade Federal do Espírito Santo, Vitória (ES), Brasil, **3** Instituto Nacional de Saúde da Mulher, da Criança e do Adolescente Fernandes Figueira, Rio de Janeiro (RJ), Brasil, **4** Universidade de São Paulo, São Paulo (SP), Brasil

☯ These authors contributed equally to this work.
* rosa.domingues@ini.fiocruz.br

## Abstract

The high fertility rate (FR) in adolescents is a global public health problem, with the Americas region having the second highest FR in the world. This study aims to carry out a temporal and spatial analysis of the FR in adolescents in Brazil in the period 2012−2021 and evaluate the effects of primary care population (PCP) coverage on the FR in adolescents in different contexts of human development, measured by the Municipal Human Development Index (HDI-M). This is an ecological study developed in three stages: 1) time trend analysis of the FR in girls aged 10–14 years and 15–19 years; 2) spatial analysis of the FR in girls aged 10–14 years and 15–19 years; and 3) analysis of the effect of PCP coverage on the FR in girls aged 10–19 years in different contexts of HDI-M, before and during the COVID-19 pandemic. In the period from 2012 to 2021, FR for girls aged 10–14 and 15–19 years had a statistically significant decrease, going from 3.4 to 2.4 and from 61.9 to 45.3 per 1,000, respectively. In the year 2021, the downward trend attenuated for girls aged 15–19 years and stopped for girls aged 10–14 years. Clusters with high FR were observed in the North and Midwest regions of the country. Higher FR were observed in municipalities with lower HDI-M and lower PCP coverage, with the increase in PCP coverage having a more intense effect on reducing the FR in municipalities with lower HDI-M. There was an attenuation of the effect of the increase in PCP coverage in the FR during the pandemic period, reinforcing the importance of guaranteeing access to sexual and reproductive health services even in contexts of health crisis.

**Data availability statement:** All data and related metadata underlying the findings are available at https://github.com/observatorioobstetrico/fertility_rates_article.

**Funding:** This work was supported by the Bill & Melinda Gates Foundation [INV-027961] and the Brazilian Ministry of Health/DECIT/CNPq [grant number 445116/2020-0]. There was no additional internal or external funding received for this study. Under the grant conditions of the Foundation, a Creative Commons Attribution 4.0 Generic License has already been assigned to the Author Accepted Manuscript version that might arise from this submission. Author who received the funding: RMSMD Name of organisation to be invoiced: Bill & Melinda Gates Foundation Address of organisation: 500 5th Ave N, Seattle, Washington 98109 Name of contact: Gates Foundation Open Access Publishing Email address: openaccess@gates-foundation.org Project number: INV-027961"

**Competing interests:** The authors have declared that no competing interests exist.

## Introduction

The reduction in the fertility rate is a global phenomenon, falling from 4.84 (95% CI 4.63–5.06) per 1,000 in 1950 to 2.23 (95% CI: 2.09–2.38) per 1,000 in 2021, with more than half of the countries presenting values below the replacement rate of 2.1 in 2021 [1]. However, this decline is heterogeneous across countries, being less pronounced in less developed countries. It is estimated that by 2100, more than half of births will occur in Sub-Saharan Africa. In Brazil, the fertility rate showed a sharp reduction, especially after the year 2000 [2], reaching a value of 1.93 (1.78–2.12) per 1,000 women of childbearing age in the year 2021 [1].

Differences in fertility rates according to age group are observed between countries with different levels of development, with the Africa and Latin America regions having the highest fertility rates in girls aged 15–19 [3,4]. Despite advances, with a drop in the fertility rate in Latin American countries, inequities persist, with indigenous or Afro-descendant people, from lower-income families, and living in rural regions, presenting a higher frequency of teenage pregnancy [4]. A study that evaluated indicators related to the Sustainable Development Goals in eight Latin American countries in the period 2010–2020 found that the fertility rate in adolescents was the one that showed the smallest drop in the period and the one that maintained the greater inequity, with rates five times higher in adolescents from poorer families [5]. In Brazil, a study evaluating the fertility rate in adolescents up to 2019 identified a drop in the fertility rate in girls aged 15–19 since 2000 and a drop in the fertility rate in girls aged 10–14 since 2010 [6].

The fertility rate in adolescents is associated with lower economic development, measured by the Human Development Index (HDI) [6–9], with income inequality, measured by the Gini index [10]; and gender inequity and female autonomy, measured by indicators of work, education and access to health [11]. In cities in Latin America, a lower rate was observed in places with greater access to health care, measured by coverage of preventive exams for cervical-uterine cancer and mammography exams [11].

In Brazil, one study found an association between a higher fertility rate in adolescents aged 15–19 and lower income, education and coverage of the family health strategy [12], while another demonstrated the effects of the cash transfer program "Bolsa Família" in reducing this rate of fertility [13]. However, no studies were identified that evaluated the effects of primary care population (PCP) coverage on the fertility rate in adolescents in different contexts of economic development.

It is also unclear what effect PCP coverage had on adolescent fertility rates during the COVID-19 pandemic in 2020 and 2021. Studies have reported that the COVID-19 pandemic has affected access to various health services, including sexual and reproductive health services [14–17], with increases in adolescent pregnancies reported in some countries [18,19].

This study aims to carry out a temporal and spatial analysis of the fertility rate in adolescents aged 10–14 and 15–19 years in the period 2012−2021, including the first

two years of the COVID-19 pandemic, as well as evaluating the effects of PCP coverage on the fertility rate in adolescents in different contexts of human development, before and during the pandemic.

## Materials and methods

### Study design

This is an ecological study developed in three stages: 1) time trend analysis of the fertility rate in girls aged 10–14 years and 15–19 years during the period 2012−2021; 2) spatial analysis of the fertility rate in girls aged 10–14 years and 15–19 years before (2018−2019) and during the COVID pandemic (2020−2021); and 3) analysis of the effect of PCP coverage on the fertility rate in girls aged 10–19 years in different contexts of human development, before (2018−2019) and during (2020−2021) the COVID-19 pandemic.

### Variables and data source

For all analysis, the fertility rate per 1,000 women was the outcome variable, being calculated using the formula: number of live births (LB) in women of a given age group in the location and period of interest divided by the female population of the same age group, location and period of interest, multiplied by 1,000. In the time series analysis, the variables year and local of LB occurrence (27 Brazilian Federative Units) were used. In the spatial analysis, the unit of analysis was the 5,570 Brazilian municipalities. To analyze the effects of PCP on the fertility rate in women under 20 years of age, the unit of analysis was also the 5,570 Brazilian municipalities and we used the variables "PCP coverage"; the "Municipal Human Development Index (HDI-M)", calculated from 2010 demographic census data; and the "year of occurrence of the LB". The LB number was obtained from the Live Birth Information System (SINASC) (https://datasus.saude.gov.br/informa-coes-de-saude-tabnet/); the number of women residing per municipality and federative unit from the Brazilian Ministry of Health based on the 2022 demographic census carried out by the Brazilian Institute of Geography and Statistics/IBGE (http://tabnet.datasus.gov.br/cgi/deftohtm.exe?ibge/cnv/popsvs2024br.def); the HDI-M from "Atlas Brasil" (http://www.atlasbrasil.org.br/acervo/atlas); and the "PCP coverage" from the Brazilian Ministry of Health (https://sisaps.saude.gov.br/painelsaps/cobertura_pot_aps).

### Data analysis

For the time series analysis, we used the Mann-Kendall non-parametric test [20] to detect trends in the time series of the fertility rate of girls aged 10–14 years and 15–19 years, from 2012 to 2021, in Brazil and in each of the 27 Brazilian Federative Units. A negative value of the statistic test indicates a downward trend, a positive value indicates an upward trend and a statistic close to zero indicates no trend, with a p value of less than 0.05 indicating statistical significance. We also described year-to-year variations in the fertility rate in the two years before (2018−2019) and during (2020−2021) the COVID-19 pandemic in Brazil.

For the spatial analysis, we analyzed the fertility rates of girls aged 10–14 years and 15–19 years in two periods: 2018–2019 (pre-pandemic) and 2020–2021 (during the pandemic). The choice of these two periods was based on our initial exploratory analyses, when we verified that the clustering of municipalities remained stable throughout the earlier years of the series (2012–2017). In other words, the inclusion of these years did not change the overall patterns of spatial distribution or the identification of municipalities with high and low fertility rates. Given this stability, we opted to present the analysis for the two years immediately prior to the pandemic (2018 and 2019) in order to ensure temporal comparability with the analysis during the pandemic period (2020 and 2021), which was also conducted using two years of data. This decision aimed to provide a more direct and balanced comparison between the pre-pandemic and pandemic scenarios, while avoiding unnecessary redundancy by including additional periods that did not change the structure of the clusters.

The focus of our clustering analysis was on grouping municipalities with similar fertility behavior, even if they are geographically distant. For this reason, we chose to use non-spatial clustering methods, as they allowed us to form groups based solely on the outcome of interest without introducing spatial constraints that could obscure relevant similarities. To identify patterns and similarities we used the following grouping methods (clusters): k-means, k-medoids, Ward's method, single linkage, complete linkage and average linkage [21]. For each clustering method, we calculated the following performance metrics: Davies-Bouldin, Dunn, silhouette and Calinski-Harabasz indices [22]. For each age group and for each period, we chose the grouping method that presented the best performance metrics. Subsequently, we carried out Dunn's multiple comparison [23] analyzes between pairs of groups of municipalities grouped by the fertility rate (per thousand) of each age group. We applied Dunn's test as an additional step to assess pairwise differences in fertility rates between clusters, providing statistical evidence to support the interpretation of the group structure. While k-means optimizes partitioning based on a distance metric, it does not, by itself, yield information on the statistical significance of the observed differences across clusters. Dunn's test, therefore, complements the clustering by offering inferential support that reinforces the descriptive patterns identified.

We generated maps, presented in Figs 2 and 3, by using the R software, specifically with the geobr and ggplot2 packages. The geobr package (https://github.com/ipeaGIT/geobr) provides official spatial data from Brazil, obtained directly from governmental sources such as the IBGE, which are in the public domain and do not have copyright restrictions for use in research and publications. The geobr package itself is licensed under the MIT License. The ggplot2 package, used for visualization and customization of these data, is also licensed under the MIT License. Therefore, the underlying map image is based on public data from the Brazilian government, accessed through a widely used and openly licensed R package, and carries no copyright restrictions for scientific publication.

To analyze the factors associated with the fertility rate in women under 20 years of age in Brazilian municipalities, the Generalized Additive Models for Location, Scale, and Shape (GAMLSS) [24] were fitted, with the adolescent fertility rate as the response variable and the covariates "HDI-M", "PCP coverage", and "year" (pre-pandemic/pandemic). The analysis period corresponds to the years 2018–2021, with 2 pre-pandemic years (2018 and 2019) and 2 pandemic years (2020 and 2021). We used the year variable as a random effect in the model to take into account the correlation of rates in the same municipality over the years. We adjusted several GAMLSS models, with the "Asymmetric t-Distribution Type 4 (ST4)" model being chosen, because it presented a lower Akaike value (AIC) and better suitability, considering the analysis of normalized quantile residues and the worm plots analysis [25]. We adjusted a full model, including interactions between covariates; a reduced model, including only the significant terms in the full model; and other reduced models, with variables selected from the full model based on stepwise selection. We used the generalized likelihood ratio test (GLR) to test the null hypothesis that the reduced model is better than the full model. We adopted a significance level of 5% and carried out all statistical analyzes using the R software [26].

SINASC has high coverage in the country, but coverage can vary especially in small municipalities. To investigate the issue of under-registration of live births, we conducted a complementary analysis using the estimates of birth registration coverage in SINASC at the municipal level from the Brazilian Institute of Geography and Statistics available for the period 2015–2021 (https://www.ibge.gov.br/estatisticas/sociais/populacao/26176-estimativa-do-sub-registro.html?edicao=32265&t=resultados). We corrected for under-registration by assuming that birth registration coverage in SINASC did not vary according to the woman's age.

All data and related metadata underlying the findings are available at https://github.com/observatorioobstetrico/fertility_rates_article. This project used de-identified publicly accessible data and is exempt from ethical consideration.

## Results

### Time trend analysis of the fertility rate in Brazil and the Federative Units

In the period from 2012 to 2021, fertility rates for girls aged 10–14 and 15–19 years old showed statistically significant decreases in Brazil, going from 3.4 to 2.4 and from 61.9 to 45.3 per 1,000, respectively. The reduction in these rates was also observed in most Federative Units, except in the age group of 10–14 years in the Federative Units of Acre, Piauí and Roraima, where they showed non-significant decreasing trends, and in the age group of 15–19 years in the Federative Unit of Roraima, where it showed a non-significant decreasing trend (decrease from 94.7 to 92.6, p value = 0.720) (Table 1).

**Table 1. Test statistics and p-value of the Mann-Kendall test for the fertility rate of the age groups 10 to 14 years and 15 to 19 years. Brazil and Federative Units, 2012 - 2021.**

| Location/age | 10-14 years old | | | | 15-19 years old | | | |
|---|---|---|---|---|---|---|---|---|
| | FR 2012 | FR 2021 | Mann-Kendall | p-value | FR 2012 | FR 2021 | Mann-Kendall | p-value |
| Brazil | 3.4 | 2.4 | −3.282 | 0.00103 | 61.9 | 45.3 | −3.399 | 0.00068 |
| Acre | 7.1 | 6.4 | −1.443 | 0.14911 | 101.3 | 79.4 | −3.143 | 0.00167 |
| Amapá | 5.8 | 5.1 | −2.525 | 0.01158 | 94.9 | 78.5 | −3.041 | 0.00236 |
| Amazonas | 7.2 | 6.2 | −3.066 | 0.00217 | 101.9 | 88.1 | −3.041 | 0.00236 |
| Pará | 5.7 | 4.9 | −2.735 | 0.00624 | 88.9 | 73.0 | −3.323 | 0.00089 |
| Rondônia | 3.6 | 2.7 | −2.326 | 0.02004 | 73.6 | 54.6 | −3.757 | 0.00017 |
| Roraima | 8.3 | 7.1 | −1.803 | 0.07133 | 94.7 | 92.6 | −0.358 | 0.72051 |
| Tocantins | 5.3 | 4.3 | −2.451 | 0.01424 | 79.8 | 65.0 | −3.220 | 0.00128 |
| Alagoas | 6.1 | 4.3 | −3.607 | 0.00031 | 81.5 | 68.5 | −2.784 | 0.00537 |
| Bahia | 4.0 | 3.0 | −3.387 | 0.00071 | 65.4 | 48.3 | −3.935 | 0.00008 |
| Ceará | 3.9 | 3.0 | −2.735 | 0.00624 | 59.7 | 45.2 | −3.220 | 0.00128 |
| Maranhão | 5.3 | 4.7 | −2.917 | 0.00353 | 84.0 | 70.2 | −2.504 | 0.01227 |
| Paraíba | 3.7 | 2.6 | −2.705 | 0.00683 | 64.8 | 53.4 | −3.041 | 0.00236 |
| Pernambuco | 4.3 | 2.9 | −3.323 | 0.00089 | 71.5 | 52.4 | −3.578 | 0.00035 |
| Piauí | 3.7 | 3.2 | −1.823 | 0.06827 | 67.8 | 55.2 | −2.504 | 0.01227 |
| Rio Grande do Norte | 4.0 | 2.6 | −3.143 | 0.00167 | 62.5 | 44.9 | −3.578 | 0.00035 |
| Sergipe | 4.1 | 3.3 | −2.525 | 0.01158 | 67.7 | 52.1 | −2.862 | 0.00421 |
| Distrito Federal | 2.1 | 1.1 | −3.099 | 0.00194 | 49.0 | 29.5 | −3.757 | 0.00017 |
| Goiás | 3.1 | 1.9 | −3.426 | 0.00061 | 59.9 | 43.3 | −3.220 | 0.00128 |
| Mato Grosso | 4.1 | 3.7 | −2.735 | 0.00624 | 73.8 | 61.6 | −2.963 | 0.00304 |
| Mato Grosso do Sul | 5.0 | 3.2 | −3.399 | 0.00068 | 78.1 | 55.9 | −3.399 | 0.00068 |
| Espírito Santo | 2.8 | 2.3 | −2.425 | 0.01532 | 57.7 | 43.2 | −3.220 | 0.00128 |
| Minas Gerais | 2.0 | 1.4 | −3.099 | 0.00194 | 48.2 | 34.9 | −3.399 | 0.00068 |
| Rio de Janeiro | 2.9 | 1.9 | −3.041 | 0.00236 | 56.6 | 40.3 | −2.963 | 0.00304 |
| São Paulo | 2.2 | 1.1 | −3.502 | 0.00046 | 51.7 | 30.8 | −3.722 | 0.00020 |
| Paraná | 3.0 | 1.6 | −3.426 | 0.00061 | 59.1 | 36.9 | −3.578 | 0.00035 |
| Rio Grande do Sul | 2.2 | 1.3 | −3.066 | 0.00217 | 50.2 | 31.2 | −3.502 | 0.00046 |
| Santa Catarina | 2.2 | 1.1 | −3.350 | 0.00081 | 48.2 | 35.0 | −2.963 | 0.00304 |

FR, fertility rate.

Fig 1 presents the time series from 2012 to 2021 of fertility rates for the age groups 10–14 years and 15–19 years in Brazil. For the 10- to 14-year-old age group, there is an average decrease of 7.69% in the fertility rate per year in the pre-pandemic period (2018 and 2019). During the pandemic, there was a decrease of 8.33% from 2019 to 2020, and the rate was stationary from 2020 to 2021. For the age group from 15 to 19 years old, the average decrease in the fertility rate per year before the pandemic was 6.55%. During the pandemic, there was a 7.92% decrease from 2019 to 2020 and a 3.09% decrease from 2020 to 2021.

**Spatial analysis of the fertility rate**

In both periods (pre-pandemic and pandemic), and in both age groups, the best grouping method was k-means with 3 groups (Table 2). In the 10–14 years age group, in the pre-pandemic period, group 1 is made up of 3,012 municipalities with an average fertility rate of 0.94 (standard deviation/SD 0.95), group 2 of 2,162 municipalities with an average fertility rate of 4.40 (SD 1.26) and group 3 by 396 municipalities with an average fertility rate of 10.34 (SD 4.28) LB per 1,000 women. During the pandemic, for this same age group, group 1 is made up of 3,274 municipalities with an average fertility rate of 0.90 (SD 0.93), group 2 is made up of 1,986 municipalities with an average fertility rate of 4.44 (SD 1.32) and group 3 by 310 municipalities with an average fertility rate of 11.14 (SD 4.24) LB per 1,000 resident women. In the 15–19 years age group, in the pre-pandemic period, group 1 is made up of 2,357 municipalities with an average fertility rate of 35.44 (SD 9.49), group 2 is made up of 2,466 municipalities with an average fertility rate of 61.25 (SD 8.57) and group 3 by 747 municipalities with an average fertility rate of 96.83 (SD 17.69) LB per 1,000 resident women. During the pandemic, for this same age group, group 1 is made up of 2,244 municipalities with an average fertility rate of 30.68 (SD 8.80), group 2 is made up of 2,499 municipalities with an average fertility rate of 54.50 (SD 8.01) and group 3 by 827 municipalities with an average fertility rate of 88.61 (SD 18.29) LB per 1,000 resident women.

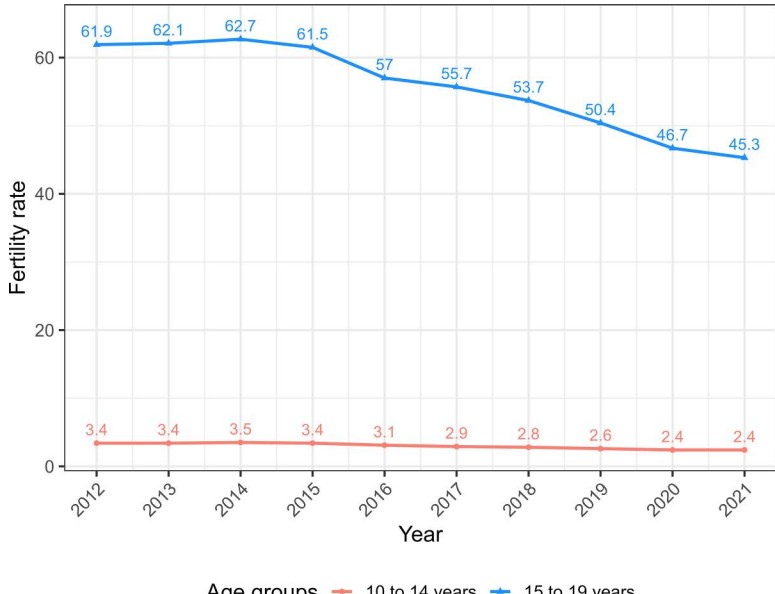

**Fig 1. Evolution of fertility rates in the age groups of 10 to 14 years and 15 to 19 years in Brazil, 2012–2021.**

**Table 2. Clustering validation metrics for fertility rates in the age groups of 10 to 14 years and 15 to 19 years. Brazil, 2018-2019 and 2020-2021.**

| Method | Davies Bouldin (centroid) | Davies Bouldin (medoid) | Dunn Index | Silhouette Index | Calinski-Harabasz (centroid) | Calinski-Harabasz (medoid) |
|---|---|---|---|---|---|---|
| **10 to 14 age group, years 2018–2019** | | | | | | |
| kmedoids3 | 0.8198007 | 1.010448 | 0.0021692 | 0.5658505 | 6140.89 | 5859.125 |
| kmedoids4 | 0.8037138 | 1.063107 | 0.0022371 | 0.5885946 | 6398.828 | 6004.266 |
| kmeans3 | 0.8336077 | 1.002882 | 0.0023148 | 0.5800877 | 7973.284 | 7776.422 |
| kmeans4 | 0.7828237 | 1.013612 | 0.0024876 | 0.5864388 | 9010.328 | 8048.691 |
| ward2 | 0.8790065 | 1.096638 | 0.0021505 | 0.6092797 | 7193.089 | 6985.303 |
| ward3 | 0.8480933 | 1.060615 | 0.0023981 | 0.5786352 | 7022.586 | 6973.459 |
| **10 to 14 age group, years 2020–2021** | | | | | | |
| kmedoids3 | 0.8172548 | 1.0793661 | 0.002770083 | 0.5656204 | 6121.451 | 5684.212 |
| kmedoids4 | 0.7682952 | 1.021208 | 0.002932551 | 0.5952804 | 7495.894 | 7063.963 |
| kmeans3 | 0.764458 | 0.9449188 | 0.003076923 | 0.5973804 | 9097.737 | 8762.034 |
| kmeans4 | 0.6547798 | 0.7422809 | 0.003472222 | 0.5978453 | 11615.918 | 10492.895 |
| ward2 | 0.8329748 | 1.029808 | 0.002816901 | 0.6397901 | 7363.678 | 7155.641 |
| ward4 | 0.6518735 | 0.7245958 | 0.003649635 | 0.5881173 | 11320.427 | 9823.375 |
| **15 to 19 age group, years 2018–2019** | | | | | | |
| kmedoids3 | 0.7557952 | 0.8824612 | 0.000692 | 0.5152591 | 9469.798 | 9049.865 |
| kmeans3 | 0.7248163 | 0.8444368 | 0.0007418 | 0.5366937 | 10245.147 | 9822.237 |
| kmeans4 | 0.7385346 | 0.875213 | 0.0008006 | 0.5165412 | 11461.608 | 10872.393 |
| ward3 | 0.7382885 | 0.8592175 | 0.000721 | 0.5320165 | 10148.539 | 9710.915 |
| **15 to 19 age group, years 2020–2021** | | | | | | |
| kmedoids3 | 0.7678656 | 0.9280219 | 0.0006131 | 0.5107424 | 8747.872 | 8189.097 |
| kmedoids4 | 0.7720505 | 0.9338004 | 0.0006452 | 0.49429 | 9215.322 | 8629.786 |
| kmeans3 | 0.7490219 | 0.8861012 | 0.0006506 | 0.5280106 | 9674.998 | 9194.743 |
| kmeans4 | 0.7205759 | 0.8296064 | 0.0007305 | 0.5311639 | 11473.176 | 11175.003 |
| ward3 | 0.7465943 | 0.8912756 | 0.0006188 | 0.5129649 | 8687.645 | 8171.134 |

Note: For the 10–14 years old groups, in the two periods, the better performance was observed for K-Means with K = 4, but some clusters contained only a few municipalities. Therefore, the chosen method was the K-Means with K = 3, since it had the best overall performance after K-Means with K = 4. For the 15–19 years old group, period 2018–2019, K-Means with K = 3 performed better in most of the metrics. For the 15–19 years old group, period 2020–2021, K-Means with K = 4 performed better, but K-Means with K = 3 was chosen to make it "comparable" with the other chosen clustering methods.

The p values of Dunn's tests for multiple comparisons between pairs of groups (three comparisons) of each age group and each period were all very close to zero, indicating significant differences between the three groups in relation to the fertility rate. For the 10–14 years age group, group 3 (municipalities with the highest fertility rates) is predominantly formed by municipalities in the North region of Brazil, while group 1 (municipalities with the lowest fertility rates) is mainly composed of municipalities in the Southeast and South regions, with a similar pattern for the two periods of analysis (Fig 2). For the 15–19 years old age group, group 3 (municipalities with the highest fertility rates) is predominantly formed by municipalities located in the North of Brazil and in the Southeast of the state of Mato Grosso. Group 1 (municipalities with the lowest fertility rates) is predominantly composed of municipalities in the Southeast and South regions (Fig 3).

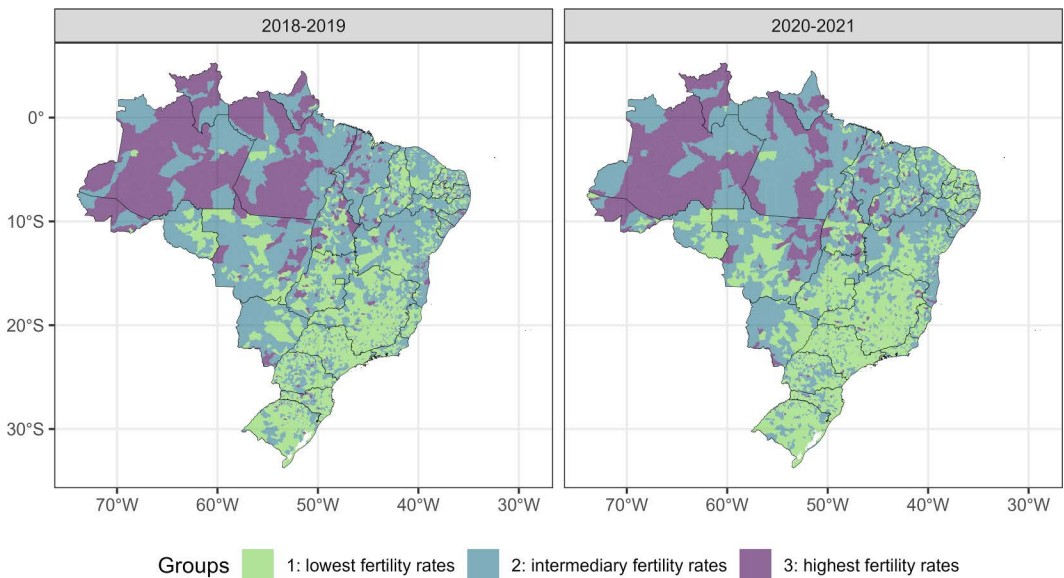

**Fig 2. Distribution of groups of municipalities formed by municipalities with similar fertility rate of women aged 10 to 14 years in the pre-pandemic period (on the left) and in the pandemic period (on the right).** Footnote: Maps generated with the use of the geobr package (https://github.com/ipeaGIT/geobr) licensed under the MIT License with no copyright restrictions for scientific publication.

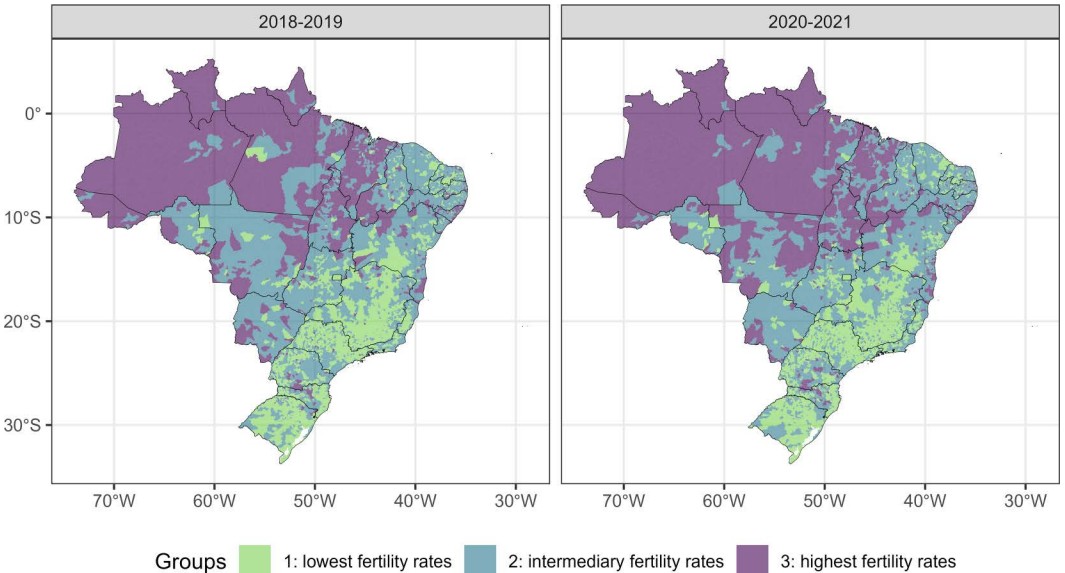

**Fig 3. Distribution of groups of municipalities formed by municipalities with similar fertility rate of women aged 15 to 19 years in the pre-pandemic period (on the left) and in the pandemic period (on the right).** Footnote: Maps generated with the use of the geobr package (https://github.com/ipeaGIT/geobr) licensed under the MIT License with no copyright restrictions for scientific publication.

### Factors associated with the fertility rate in women under 20 years of age

The reduced model chosen by the second stepwise strategy had the best adjustment, with the following results for the period 2018−2021: 1) individually, being a year of the COVID-19 pandemic generates a decrease in the value of the

fertility rate in women under 20 years of age in the municipality; 2) individually, the increase in PCP coverage generates a decrease in the value of the fertility rate in women under 20 years of age in the municipality; 3) individually, the increase in the HDI-M generates a decrease in the value of the fertility rate in women under 20 years of age in the municipality; 4) for years of the COVID-19 pandemic, the increase in PCP coverage generates an increase in the value of the fertility rate in women under 20 years of age in the municipality; 5) the joint increase in the HDI-M and the PCP coverage generates an increase in the value of the municipality's fertility rate (Table 3).

In Fig 4, the results of the effect of increasing the PCP coverage on the fertility rate in women under 20 years of age are presented in three HDI-M scenarios: quantile 0.25, median value and quantile 0.75 among all municipalities, resulting in HDI-M values of, respectively, 0.599, 0.665 and 0.718 in the corresponding period. We observed that:

1.  For a municipality with HDI-M in quantile 0.25 (0.599), an increase of 10 units in PCP coverage generates:

a)  For the pre-pandemic years, an average decrease of 1.553 units in the fertility rate of women under 20 years of age in the municipality;

b)  For pandemic years, an average decrease of 1.322 units in the fertility rate of women under 20 years of age in the municipality

2)  For a municipality with a median HDI-M (0.665), an increase of 10 units in PCP coverage generates:

a)  For pre-pandemic years, an average decrease of 1.069 units in the fertility rate of women under 20 years of age in the municipality;

b)  For pandemic years, an average decrease of 0.838 units in the fertility rate of women under 20 years of age in the municipality;

3)  For a municipality with HDI-M in quantile 0.75 (0.718), an increase of 10 units in PCP coverage generates:

a)  For pre-pandemic years, an average decrease of 0.68 units in the fertility rate of women under 20 years of age in the municipality;

b)  For pandemic years, an average decrease of 0.449 units in the fertility rate of women under 20 years of age in the municipality.

For the entire period 2018–2021, an increase of 10 units in the PCP coverage in a municipality with HDI-M in quantile 0.25 generates a decrease of 1.438 units in the fertility rate of women under 20 years of age. For a municipality with a median HDI-M, this increase of 10 units in PCP coverage generates a decrease of 0.953 units in the fertility rate of women under 20 years of age. Finally, in a municipality with HDI-M in quantile 0.75, an increase of 10 units in the PCP coverage generates a decrease of 0.565 units in the fertility rate of women under 20 years of age in the municipality.

**Table 3. Adjusted model of the effect of primary care coverage, the Municipal Human Development index and the COVID-19 pandemic on the fertility rate of women under 20 years of age. Brazil, 2018-2021.**

| Variable | Estimate | standard error | t-statistic | p-value |
|---|---|---|---|---|
| Intercept | 138.418 | 4.25 | 32.57 | <0.0001 |
| Pandemic year: yes | −4.910 | 0.55 | −8.92 | <0.0001 |
| PCP coverage | −0.595 | 0.04 | −13.17 | <0.0001 |
| HDI-M | −150.472 | 5.66 | −26.56 | <0.0001 |
| Pandemic year: PCP coverage | 0.023 | 0.01 | 3.64 | <0.0001 |
| PCP coverage: HDI-M | 0.734 | 0.06 | 12.07 | <0.0001 |

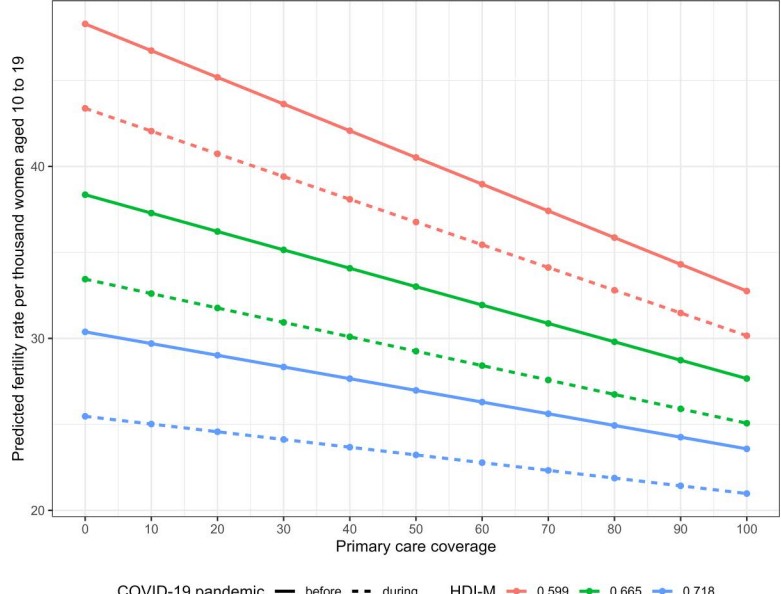

**Fig 4. Predicted values of the fertility rate of women under 20 years of age according to HDI-M values, Primary Care Population coverage and information on the year of the pandemic.** Brazil, 2018-2021.

## Complementary analysis on under-registration in SINASC

During the period from 2012 to 2021, the average number of municipalities with zero births per year was approximately 47 and no municipality recorded zero births throughout the entire period from 2012 to 2021. This indicates that while some municipalities reported zero births in specific years (often small municipalities), the phenomenon was not persistent across time. Therefore, these cases likely reflect natural demographic variability in small populations rather than systematic underreporting or data anomalies.

The lowest coverage of SINASC was observed in Condor (Rio Grande do Sul), with 24% in 2015. However, the vast majority of municipalities exhibited very high coverage, with the first quartile at 98.4%. We redid the analyses using under-registration correction. The macro-level patterns remained unchanged and the new fitting attempts showed minimal influence on the results — in the case of modeling, for instance, the differences in the estimated coefficients were at most 5%, and the differences in the interpretations obtained were at most 2%. For these reasons, and because data on SINASC coverage are only available from 2015 onward, preventing their use in the time trend analysis covering the entire period from 2012 to 2021, we opted to calculate the fertility rates in their raw form, without applying additional correction techniques. The complete analysis is available at https://github.com/observatorioobstetrico/fertility_rates_article/blob/main/supplementary_material2.pdf.

## Discussion

The results of this study show a reduction in the fertility rate in girls aged 10–14 and 15–19 years in Brazil during the period 2012–2021, with an attenuation of the downward trend in girls aged 15–19 years and interruption of fall in girls aged 10–14 years in the year 2021. In girls aged 10–14, clusters with higher fertility rates were observed mainly in municipalities in the North region and, in girls aged 15–19 years, in municipalities in the North and southeast regions of the Central-West region of the country. In both age groups, there was a reduction in the number of municipalities in the group with the highest fertility rate during the two years of the pandemic, which may be a consequence of the downward trend

in the fertility rate that was already occurring before the pandemic. Higher fertility rates in women under 20 years of age were observed in municipalities with lower HDI-M and lower PCP coverage, with the effect of increasing PCP coverage on reducing the fertility rate being more pronounced in municipalities with lower HDI-M. An interaction was verified between PCP coverage and the period of analysis (pre-pandemic and pandemic), with a smaller effect of PCP coverage on the fertility rate during the pandemic.

Adolescent pregnancies are associated with a greater occurrence of negative maternal and perinatal outcomes [27,28], having long-term social and economic effects, due to the interruption of studies [29,30], fewer work opportunities and the perpetuation of intergenerational cycles of poverty [3,31]. Latin America has the second highest global fertility rate in adolescents, and reducing this rate is one of the Sustainable Development Goals in the Americas region [32].

The observed reduction in the fertility rate in the 15–19 age group is consistent with previous studies, which report a reduction in this rate in the country since the year 2000 [6,10]. Reduction in the fertility rate in girls aged 15–19 has also been reported in nine Latin American countries with data available after 2010 [4]. For fertility rates in the 10–14 years age group, a national study reported a reduction in the rate in all regions and Federative Units of the country in the period 2010–2019, except in the Federative Units of Amazonas and Maranhão [6].

In this study, the only Federative Units that did not show a significant reduction in the fertility rate in girls aged 10–14 were Acre, Piauí and Roraima, while Roraima showed a non-significant decrease in the fertility rate of women aged 15–19 years. Roraima is a Federative Unit located on the border with Venezuela, having experienced an intense population immigration, with an estimated immigration of almost 300 thousand Venezuelans in the period 2017–2019 [33]. It is possible that the non-significant decrease in the fertility rate is due to a greater number of live births of immigrant women, without these women having been included in the denominator, even though we have used population estimates from the 2022 census.

The non-significant reduction in the fertility rate in girls aged 10–14 years observed in three Federative Units may include other determinants. In Brazil, sexual relations with girls aged 10–13 are classified as the result of sexual violence (presumed rape). A study that calculated the date of conception of live births of 14-year-old girls estimated that in about 60% of these cases the sexual intercourse that resulted in pregnancy occurred at the age of 13 [34]. Therefore, the vast majority of pregnancies in girls aged 10–14 in the country could be the result of sexual violence and their non-significant reduction may reflect local contexts of greater vulnerability. A previous nationwide study, carried out in the period 2000–2012, identified a higher fertility rate among 10–14 years-old girls in border regions and tourist areas [7]. Conflicts in mining areas and social and cultural aspects may also be relevant in the North region, since it has the highest proportion of indigenous women and of population living in rural regions where early pregnancy may have a different meaning, despite the legal definition. The percentage of live births among indigenous girls aged 10–14 is almost four times higher than among white girls [35]. Future studies should investigate local aspects that may be related to the high fertility rate among 10–14 years-old girls in specific contexts.

The reduction in the downward trend in the fertility rate in girls aged 15–19 years and the interruption of the decline in girls aged 10–14 years in 2021 is consistent with studies that report an increase in teenage pregnancy during the COVID-19 pandemic [18,19]. Explanations for the effects of the COVID-19 pandemic on the sexual and reproductive health of adolescents in low- and middle-income countries include limited access to sexual and reproductive health services, school closures, increases in early marriages, sexual or intimate partner violence during the pandemic, discontinuity of maternity services and involvement of adolescents in risky or exploitative work [36]. In Brazil, a study that analyzed the fertility rate in adolescents in 2020, compared to 2019, found an 8.4% reduction, which the authors considered to be an unexpected result [37], given the report of an increase in teenage pregnancies in other countries. However, pregnancies in 2020 largely reflect the standard of care that existed in 2019 and early 2020, before COVID-19 was declared a pandemic. More evident effects of the pandemic on the fertility rate, such as those described in this study, would be expected in 2021, reflecting the change in context that occurred in 2020.

The higher fertility rate in women under 20 years of age in municipalities with lower HDI-M and lower PCP coverage is consistent with previous studies that show an association between the fertility rate in adolescents and lower economic development [6–9], income inequities [4,5,10], and lower access to health services [11,12]. Innovatively, our results show a differentiated effect of increasing PCP coverage on the fertility rate in women under 20 years of age in municipalities with the lowest HDI-M, which are the most vulnerable. A smaller effect was observed during the pandemic period, that is, the ability of the PCP coverage to affect the decline in the fertility rate was attenuated during the pandemic, which may reflect less specific access to sexual and reproductive health services, for the same global PCP coverage.

In municipalities with a higher HDI-M, we observed a smaller effect of the increase in PCP coverage in reducing the fertility rate in women under 20 years of age in both the pre-pandemic and pandemic periods. These results probably reflect the lower dependence of the population of these municipalities on public primary care services. Coverage with health plans in Brazil is associated with better economic conditions [38,39], with the proportion of users of the public sector being inversely associated with the HDI-M (correlation of −0.84). In other words, in municipalities with a higher HDI-M, lower PCP coverage does not mean a lack of access to health services, as the population that has medical health plans has access to private services, with less evident effects of the increase in PCP coverage on the fertility rates.

Primary health care has been the gateway to the Brazilian public health system since 1994, with an estimated national population coverage of 75%, with the highest coverage observed in the Northeast region and the lowest in the Southeast region. Primary health care is associated with different positive health outcomes [40–43]. Its positive effect on reducing the fertility rate in adolescents, especially in places with lower HDI-M, is further evidence of the benefits of this care model. Our study analyzed the fertility rates in the period 2012–2021, but reductions in the fertility rate in adolescents aged 15–19 and 10–14 have been reported since 2000 and 2010 [6,10], respectively, which may reflect the effects of primary health care in the country since the late 1990s. Compared to other Latin American countries with data available after 2010 [4], Brazil has the lowest fertility rate among adolescents aged 15–19, as well as the highest annual reduction rate (except for that observed in Ecuador in the period 2012–2018). The expansion of PCP coverage in the country may be one of the possible explanations for these results.

This study has some limitations. We used administrative databases and the data is subject to coverage and completion quality problems. SINASC has coverage greater than 90% in all Brazilian Federative Units [44], the coverage parameter used by the Brazilian Ministry of Health amid the Health Surveillance Actions Qualification Program [45]. However, municipal coverage varies regionally, being lower mainly in municipalities in the North and Northeast region [46]. We performed additional analyses using correction for underreporting during the period 2015–2021, and the effects on the model results were small. Data on underreporting for the years 2012–2014 are not available, and it is possible that the adolescent fertility rate, which is already higher in the North and Northeast regions, is underestimated. However, the downward trend in the fertility rate would not be affected. The HDI-M, used for the human development classification of municipalities, is calculated based on data from the 2010 census, and it is possible that changes may have occurred throughout the period analyzed. For PCP coverage in the country, we used the estimate provided by the Brazilian Ministry of Health, based on the number of professionals registered in the National Registry of Health Establishments and on the estimated population of each municipality. Although subject to limitations of these sources of information, the PCP coverage is the official Brazilian data, used for the planning and formulating of public policies. Finally, it was not possible to analyze the effects of increasing PCP coverage in the age groups of 10–14 years and 15–19 years separately due to the small number of live births. However, it is possible to assume that the observed benefits of increasing PCP coverage in reducing the fertility rate are common to both age groups, due to greater access to sexual and reproductive care services, including contraceptive methods and legal termination of pregnancy.

As strengths of the study, we highlight the use of GAMLSS models in statistical analysis. This class of models allows the choice between a wide range of distributions for the response variable, and its main advantage is the ability to separately model the parameters of location (related to the mean of the distribution), scale (related to the variance of the

distribution) and shape (related to the asymmetry and kurtosis of the distribution), bringing important flexibility to the model. The chosen model (Asymmetric T-Distribution Type 4-ST4) allows the modeling of all mentioned parameters, offering flexibility to capture different characteristics of the data distribution [47].

## Conclusion

The results of this study show a reduction in the fertility rate in girls aged 10–14 years and 15–19 years, in Brazil, in the period 2012−2021, with a reduction in the downward trend in 2021, the second year of the COVID-19 pandemic. Higher rates were observed in municipalities with lower HDI-M and lower PCP coverage, with the increase in PCP coverage having a more intense effect on reducing the fertility rate in municipalities with lower HDI-M. There was an interaction between PCP coverage and the pandemic period, with attenuation of the effect of increased PCP coverage on the fertility rate in the pandemic period, reinforcing the importance of guaranteeing access to sexual and reproductive health services even in contexts of health crisis.

## Author contributions

**Conceptualization:** Rosa Maria Soares Madeira Domingues, Agatha Sacramento Rodrigues.

**Data curation:** Rafael Sant´Ana Herzog, Agatha Sacramento Rodrigues.

**Formal analysis:** Rafael Sant´Ana Herzog, Agatha Sacramento Rodrigues.

**Funding acquisition:** Rosa Maria Soares Madeira Domingues.

**Methodology:** Rosa Maria Soares Madeira Domingues.

**Validation:** Marcos Augusto Bastos Dias, Rossana Pulcineli Vieira Francisco.

**Visualization:** Rafael Sant´Ana Herzog, Agatha Sacramento Rodrigues.

**Writing – original draft:** Rosa Maria Soares Madeira Domingues, Rafael Sant´Ana Herzog, Agatha Sacramento Rodrigues.

**Writing – review & editing:** Marcos Augusto Bastos Dias, Rossana Pulcineli Vieira Francisco.

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
