## [Decision Letter · Decision Letter 0]

PONE-D-24-43130Fertility rate among adolescents in Brazil in the period 2012-2021: effects of human development, primary care coverage and the COVID-19 pandemic.PLOS ONE

Dear Dr. Domingues,

Thank you for submitting your manuscript to PLOS ONE. After careful consideration, we feel that it has merit but does not fully meet PLOS ONE’s publication criteria as it currently stands. Therefore, we invite you to submit a revised version of the manuscript that addresses the points raised during the review process.

The reviews show the need for thorough justification of methodological choices. All Reviewers highlighted this in different ways. Reviewer 2 details requirements for methods. Reviewer 3 also highlights that further discussion would be welcome.  I am very sorry the first round took so much time, but securing relevant reviewers during the period was very difficult. Fortunately the Authors are now able to receive detailed reviews for appreciation.

We look forward to receiving your revised manuscript.

Kind regards,

Claudia Garcia Serpa Osorio-de-Castro, Ph.D

Academic Editor

PLOS ONE

Journal Requirements:

“This work was supported by the Bill & Melinda Gates Foundation [INV-027961] and the Brazilian Ministry of Health/DECIT/CNPq [grant number 445116/2020-0]. Under the grant conditions of the Foundation, a Creative Commons Attribution 4.0 Generic License has already been assigned to the Author Accepted Manuscript version that might arise from this submission.

Author who received the funding: RMSMD

Name of organisation to be invoiced: Bill & Melinda Gates Foundation

Address of organisation: 500 5th Ave N, Seattle, Washington 98109

Name of contact: Gates Foundation Open Access Publishing

Email address: openaccess@gatesfoundation.org

Project number: INV-027961”

3. We note that Figures 2-3 in your submission contain [map/satellite] images which may be copyrighted. All PLOS content is published under the Creative Commons Attribution License (CC BY 4.0), which means that the manuscript, images, and Supporting Information files will be freely available online, and any third party is permitted to access, download, copy, distribute, and use these materials in any way, even commercially, with proper attribution. For these reasons, we cannot publish previously copyrighted maps or satellite images created using proprietary data, such as Google software (Google Maps, Street View, and Earth). For more information, see our copyright guidelines: http://journals.plos.org/plosone/s/licenses-and-copyright

a. You may seek permission from the original copyright holder of Figures 2-3 to publish the content specifically under the CC BY 4.0 license.

Please upload the completed Content Permission Form or other proof of granted permissions as an "Other" file with your submission

Reviewers' comments:

Reviewer's Responses to Questions

**Comments to the Author**

1. Is the manuscript technically sound, and do the data support the conclusions?

Reviewer #1: Yes

Reviewer #2: Yes

Reviewer #3: Yes

2. Has the statistical analysis been performed appropriately and rigorously? 

Reviewer #1: Yes

Reviewer #2: I Don't Know

Reviewer #3: No

3. Have the authors made all data underlying the findings in their manuscript fully available?

Reviewer #1: Yes

Reviewer #2: Yes

Reviewer #3: Yes

4. Is the manuscript presented in an intelligible fashion and written in standard English?

Reviewer #1: Yes

Reviewer #2: Yes

Reviewer #3: Yes

5. Review Comments to the Author

Reviewer #1: Meus comentário estão em anexo. De modo geral artigo é muito bem escrito e traz dados inovadores no estudo da fecundidade de 10 a 19 anos quando para além da análise temporal faz a inserção do período pandêmico e inova no uso do método. A análise espacial e seus resultados são bem interessantes. Meus comentários são a fim de esclarecer alguns aspectos que acredito serem relevantes.

Reviewer #2: This paper addresses an crucial public health matter for brazil and other nations of poor or medium development. Analytical procedures advances the scientific literature available in the field.

Report is quite well described and addresses the main results.

As for minor revisions I suggest but not condition changes in the title, from:

Fertility rate among adolescents in Brazil in the period 2012-2021: effects of human development, primary care coverage and the COVID-19 pandemic.

to: Effects of human development, primary care coverage and the COVID-19 pandemic on the fertility rate among adolescents in Brazil in the period 2012-2021

Other considerations follow bellow

Introduction:

Line 49  reduction repeat 2x, please rephrase

Lines 59-60  And/Or is not well applied, please adjust

Lines 77-78 are quite not conected with the overal rationale presented, I advise authors to better locate the importance of this argument in the context of their rationale. Restate.

Methods:

In this section, the trend analysis does not show in abstract and in the type of study in the 1st paragraph. This adjustment is necessary.

Página 13, linhas 27-28 this information only appears at the end of the Methods section, and its complementary to the one in the introdcution which was quite disconected to the rationale. Please restate the comparison made with de pre and pandemic years justifying this analytical procedure in both sections.

It is not quite clear in the explanation, why for the spacial analysis only 2 periods were selected, excluding all data form 2012 to 2017. Wouldn´t it be more appropriate to have a period as well in the beggining of the series? Why did the authors made this decision?

Moreover, spacial analysis details requires more thorough explanation and references.

Results:

Figure 2 and Figure 3 have diferences in the purple (high FR) gradient and this should be better explained in the text.

Text with results of Figure 4, page 20 lines 42-60 are not in a very well comprehensive and usual format. Authors should review this presentation.

Discussion:

Page 22 lines 35-37 mention an interaction analysis that was not mentioned in Methods analytical procedures.

Authors must include that dispite of the forecasts isolation during the pandemic might have had an effect on the adolescent opportunities of sexual intercourse as described in previous studies, although might have increased sexual violence.

Authors were very discrete in discussing the the observed benefits of increasing

PCP coverage in reducing the fertility rate, common to both age groups and this need more consideration, since it is a differencial.

Other issue is the political scenario and program change towards sexual and reproductive health of the country in face of these results. Aren´t they controversial?

Authors would advance if in this section they compare Brazil´s adolesct FR reduction to other countries, since we are following behing. PHC is up since 1994, why the effect is just in the last decade?

Reviewer #3: Adolescent Pregnancy and Public Health

Adolescent pregnancy is a significant public health issue, as complications during pregnancy and childbirth represent the leading cause of mortality among girls aged 15 to 19 years. Furthermore, its social impact is substantial, frequently leading to school dropouts. Given that its incidence is higher among the most socioeconomically vulnerable groups, adolescent pregnancy contributes to the perpetuation of poverty.

Socioeconomic and cultural factors strongly influence reproductive behavior during adolescence. Brazil has experienced a rapid decline in fertility rates; however, significant regional and socioeconomic disparities persist. These variations and differences in access to primary healthcare and education make the country a valuable case for analyzing this phenomenon. I acknowledge the relevance of the topic addressed in the article under review. However, several methodological issues require further discussion.

Text Revision and Introduction

One key aspect that deserves attention is the need for a text revision, particularly in the introduction. For instance, on page 3, line 58, the authors state: "Despite advances, with a drop in the fertility rate in some Latin American countries, inequities persist." I believe the word "some" could be removed, as all countries in the region have experienced a decline in fertility rates.

Methodology

An important issue that requires attention is the population data used to calculate fertility rates. The authors did not use the most recent population estimates (http://tabnet.datasus.gov.br/cgi/deftohtm.exe?ibge/cnv/popsvs2024br.def). Since many analyses were conducted at the municipal level, population changes may have a non-negligible impact on fertility rate calculations.

The first step of the analysis consists of assessing the trend in adolescent fertility rates for the 10–14 and 15–19 age groups from 2012 to 2021 by federative unit. To do so, the authors applied the nonparametric Mann-Kendall test to verify the existence of a monotonic increasing or decreasing trend over time. The approach is straightforward and clear.

In the spatial analysis, both age groups were again analyzed separately, but the period was restricted to 2018–2021. The authors justify this decision by the need for better comparability, ensuring that the pre-COVID-19 period (2018–2019) has the same duration as the pandemic period (2020–2021).

For this analysis, the authors used different clustering methods (k-means, k-medoids, Ward’s method, single linkage, and average linkage). They selected the best approach based on performance metrics (Davies-Bouldin, Dunn, Silhouette, and the Calinski-Harabasz index). However, the results of these metrics were not presented, and the authors merely stated that k-means with three groups were the chosen method. I consider these results to be included and discussed in the article.

Additionally, it would be essential to justify the choice of these methods, given that they do not account for spatial correlation. Spatial correlation is a relevant factor in adolescent fertility as it involves cultural aspects associated with the place of residence.

Another questionable aspect is Dunn’s multiple comparison test, given that k-means inherently aims to minimize within-cluster variance. The necessity of this additional test is not evident and should be better justified in the text.

Main Analysis

My main criticism of the article relates to the core analysis, which investigates the association between adolescent fertility rates (ages 10–19) and primary healthcare coverage at the municipal level. Although the authors have made their data and code available, ensuring transparency in their analyses, this does not exempt them from explicitly stating in the text which dependent variable was modeled.

The square root of fertility rates was used as the response variable, possibly to approximate a normal distribution. However, it is unclear why this transformation was applied instead of modeling the fertility rate directly using a Poisson or Negative Binomial model, which would be more appropriate for dealing with count data. The decision to use the square root transformation should be justified, especially considering that other approaches are more suitable for counting data.

Furthermore, the authors’ decision to consider the year as the level-2 unit in the multilevel model raises concerns. In this structure, municipalities are nested within years of pregnancy occurrence, which does not seem to reflect the data's hierarchical structure adequately.

I think the most appropriate approach would be to consider year level 1 and municipality level 2 since fertility rates within the same municipality tend to be correlated over time. Even if the study’s primary interest is comparing pandemic and pre-pandemic years, it isn't easy to justify the assumption that fertility rates within a given municipality are independent over time. This methodological decision should be revised or, at the very least, better justified in the text.

6. PLOS authors have the option to publish the peer review history of their article (what does this mean? ). If published, this will include your full peer review and any attached files.

**Do you want your identity to be public for this peer review?** For information about this choice, including consent withdrawal, please see our Privacy Policy .

Reviewer #1: No

Reviewer #2: No

Reviewer #3: No

---

## [Author Response · Author response to Decision Letter 1]

27 May 2025

Dear Editor,

We would like to thank you and the reviewers for the comments and recommendations on our manuscript “PONE-D-24-43130 Fertility rate among adolescents in Brazil in the period 2012-2021: effects of human development, primary care coverage and the COVID-19 pandemic”.

Considering the recommendation to update the population under 20 years of age, which was not available when we made the first submission, we redid all the analyses, with the changes in the methods, results and discussion highlighted in the text.

Following the suggestion of Reviewer 1, to evaluate the under-registration of births in SINASC, we performed complementary analyses considering the under-registration estimates made available by the Brazilian Institute of Geography and Statistics (IBGE) for the period 2015-2020. Since there was no significant change in the results (presented in supplementary material) and under-registration estimates are not available for the entire analysis period, we chose to maintain the analyses using the raw database.

We have include the statement “There was no additional internal or external funding received for this study” in our updated Funding Statement in the cover letter. We also changed our data availability statement of the submission form, indicating that the data is already available.

We would like to inform you that the maps in Figures 2 and 3 were generated by us using the R software, specifically with the geobr and ggplot2 packages. The geobr package provides official spatial data from Brazil, obtained directly from governmental sources such as the IBGE, which are in the public domain and do not have copyright restrictions for use in research and publications. The geobr package itself is licensed under the MIT License. The ggplot2 package, used for visualization and customization of these data, is also licensed under the MIT License. Therefore, the underlying map image is based on public data from the Brazilian government, accessed through a widely used and openly licensed R package, and carries no copyright restrictions for scientific publication. We included this explanation in the methods section and in the footnote of figures 2 and 3.

Changes to the revised text are highlighted in yellow. If we did not follow the reviewer's recommendation, the justification is presented in the responses below.

We hope we have attended all the recommendations and we are available for further clarification if necessary.

Sincerely,

Rosa Domingues

Corresponding author

Reviewer 1

Reviewer #1: Meus comentário estão em anexo. De modo geral artigo é muito bem escrito e traz dados inovadores no estudo da fecundidade de 10 a 19 anos quando para além da análise temporal faz a inserção do período pandêmico e inova no uso do método. A análise espacial e seus resultados são bem interessantes. Meus comentários são a fim de esclarecer alguns aspectos que acredito serem relevantes.

Na minha visão que o artigo tenha um problema e/ou uma limitação metodológica importante que precisa ser destacada pelos autores.

No texto na descrição acerca da estimação da Taxa de Fecundidade adolescente está assim” the fertility rate per 1,000 women was the outcome variable, being calculated using the formula: number of live births (LB) in women of a given age group in the location and period of interest divided by the female population of the same age group, location and period of interest, multiplied by 1,000.” Para o cálculo desse indicador, central para o estudo, destaco dois pontos que merecem melhor detalhamento e importância no texto.

1) É necessário que os autores indiquem os problemas que devem ser discutidos no tanto no que diz respeito ao numerados quanto ao denominador. O número de nascimentos sofre do problema de subregistro, especialmente para as regiões Norte e Nordeste. Como os autores lidaram com esse problema? O número de meninas de 10 a 14 e de 15 a 19, por serem oriundos da projeção também apresenta problemas, em especial depois da divulgação dos resultados do Censo de 2022 e a retroprojeção divulgada pelo IBGE, que corrigiu as estimativas passadas. Esses números estavam muito sobrerepresentados para a maior parte do país, o que tornava as taxas pouco consistentes em todos os casos. É o que os autores mostram para o estado de Roraima, em que apontam para a elevada imigração de Venezuelanos. Essas informações fazem toda diferença na hora da comparabilidade com outros estudos já existentes.

R: We thank the reviewer for this important observation.

In the current version of the manuscript, we have corrected the denominator using the 2022 Census data, thus addressing the concern about the overestimation of the number of girls aged 10 to 14 and 15 to 19.

Regarding the issue of under-registration of live births, we conducted a complementary analysis using IBGE’s estimates of SINASC coverage at the municipal level. This analysis is presented in the supplementary material.

Below, we summarize the SINASC coverage statistics presented by IBGE for Brazilian municipalities from 2015 to 2021:

Min 1st Qu. Median Mean 3rd Qu. Max SD

24 98.4 99.7 98.68 100 100 2.75

The lowest coverage was observed in Condor (Rio Grande do Sul), with 24% in 2015. However, the vast majority of municipalities exhibited very high coverage, with the first quartile at 98.4%.

Although this analysis was performed, we chose not to include it in the main manuscript, but rather in the supplementary material, for the following reasons:

1. Data on SINASC coverage are only available from 2015 onward, preventing their use in temporal analyses covering the entire period from 2012 to 2021;

2. Even in analyses where the correction could be consistently applied (such as clustering and modeling), new fitting attempts showed minimal influence on the results — in the case of modeling, for instance, the differences in the estimated coefficients were at most 5%, and the differences in the interpretations obtained were at most 2%.

For these reasons, and considering that the macro-level patterns remained unchanged, we opted to calculate the fertility rates in their raw form, without applying additional correction techniques.

2) Ainda sobre a estimativa das taxas de fecundidade, é necessário que os autores indiquem a metodologia adotada para a estimativa nos municípios. Isso porque diversos estudos apontam para a inviabilidade de usar os dados diretos, uma vez que se tratam de pequenos números e estes podem sofrer diversos efeitos que prejudicam as análises. O que foram feitos para os casos que em um determinado ano não havia registro de nascimento para meninas nos grupos analisados? Ou os números eram muito baixos? Será que esse dado era de fato o real ou sofreu o efeito de algum evento atípico, subenumeração, etc.? A estimativa para pequenas áreas merece maior detalhamento.

R: We appreciate the reviewer’s comment regarding the estimation of fertility rates in small areas and the challenges related to small numbers, underreporting, and potential atypical events.

To address this, we analyzed the number of municipalities that reported zero births among girls aged 10 to 19 in each year of the study period (2012–2021). The table below summarizes the annual counts of such municipalities:

Year Municipalities with zero births (age 10–19)

2012 36

2013 40

2014 39

2015 34

2016 43

2017 52

2018 39

2019 58

2020 59

2021 76

● The average number of municipalities with zero births per year was approximately 47.

● No municipality recorded zero births throughout the entire period from 2012 to 2021.

This indicates that while some municipalities reported zero births in specific years (often small municipalities), the phenomenon was not persistent across time. Therefore, these cases likely reflect natural demographic variability in small populations rather than systematic underreporting or data anomalies.

We included a new topic (complementary analysis on under-reporting in SINASC) in the results section to present these analyses and main results. The complete analyses is available at https://github.com/observatorioobstetrico/fertility_rates_article/blob/main/supplementary_material2.pdf.

3) Sobre os dados para Roraima, acho que também cabe a discussão da população indígena e o aumento do número de garimpeiros e conflitos na região, que contribuíram para o aumento dos casos de violência sexual contra meninas. Como vocês teriam os dados municipais, seria interessante mostrar as diferenças daqueles de fronteira. De modo geral achei a questão dos municípios pouco explorada nos dados, tinha a expectativa de ver mais citações de alguns exemplos com a análise de alguns municípios em destaque. Acho que isso potencializaria os resultados do paper.

R: We thank the reviewer for this observation. However, the aims of this study is to carry out a temporal and spatial analysis of the fertility rate in adolescents aged 10 to 14 and 15 to 19 years in the period 2012-2021, including the first two years of the COVID-19 pandemic, as well as evaluating the effects of PCP coverage on the fertility rate in adolescents in different contexts of human development, before and during the pandemic.Therefore, analyses of specific municipalities are beyond the scope of the manuscript. Moreover, as this is an ecological study, it only allows us to raise hypotheses to be investigated in future studies. The issues of violence on the borders and the indigenous population had already been raised in previous studies and addressed in our discussion. We expanded the discussion and included new bibliographical references.

Reviewer 2

This paper addresses an crucial public health matter for brazil and other nations of poor or medium development. Analytical procedures advances the scientific literature available in the field. Report is quite well described and addresses the main results.

As for minor revisions I suggest but not condition changes in the title, from:

Fertility rate among adolescents in Brazil in the period 2012-2021: effects of human development, primary care coverage and the COVID-19 pandemic.

to: Effects of human development, primary care coverage and the COVID-19 pandemic on the fertility rate among adolescents in Brazil in the period 2012-2021

Other considerations follow bellow

Introduction:

Line 49  reduction repeat 2x, please rephrase

Lines 59-60  And/Or is not well applied, please adjust

Lines 77-78 are quite not conected with the overal rationale presented, I advise authors to better locate the importance of this argument in the context of their rationale. Restate.

Methods:

In this section, the trend analysis does not show in abstract and in the type of study in the 1st paragraph. This adjustment is necessary. Página 13, linhas 27-28 this information only appears at the end of the Methods section, and its complementary to the one in the introdcution which was quite disconected to the rationale. Please restate the comparison made with de pre and pandemic years justifying this analytical procedure in both sections.

It is not quite clear in the explanation, why for the spacial analysis only 2 periods were selected, excluding all data form 2012 to 2017. Wouldn´t it be more appropriate to have a period as well in the beggining of the series? Why did the authors made this decision? Moreover, spacial analysis details requires more thorough explanation and references.

R: We thank the reviewer for the suggestions.

We changed the title and corrected the text in the introduction section as recommended. We included the trend analysis in the abstract and in the study design in the methods section.

Methods section: In our initial exploratory analyses, we verified that the clustering of municipalities remained stable throughout the earlier years of the series (2012 to 2017). That is, including these years did not alter the overall patterns of spatial distribution or the identification of high- and low-fertility municipalities.

Given this stability, we opted to present the analysis for the two years immediately preceding the pandemic (2018 and 2019) in order to ensure temporal comparability with the analysis during the pandemic period (2020 and 2021), which was also conducted using two years of data.

This decision aimed to provide a more direct and balanced comparison between the pre-pandemic and pandemic scenarios, while avoiding unnecessary redundancy from including additional periods that did not alter the cluster structure.

This explanation and further clarifications and references related to the spatial analysis methods used — including clustering criteria, spatial weight definitions, and software employed — have been added to the revised version of the manuscript, as suggested.

Results:

Figure 2 and Figure 3 have diferences in the purple (high FR) gradient and this should be better explained in the text. Text with results of Figure 4, page 20 lines 42-60 are not in a very well comprehensive and usual format. Authors should review this presentation.

R: Thank you for your comment but we reviewed figures 1 and 2 and there are no differences in the purple (high FR).

Figure 4 may not be in an usual format, but we believe it contains all the data necessary for its interpretation. As stated in the figure caption, the different colors represent municipalities with HDI-M of 0.599, 0.655 and 0.718, while the solid and dotted lines represent the pre-pandemic and pandemic periods, respectively. In the text, we present the average reduction in the fertility rate for each combination of HDI-M value and period. We would like to point out that the other reviewers did not complain about this presentation, so we decided to keep it in its current format.

Discussion:

Page 22 lines 35-37 mention an interaction analysis that was not mentioned in Methods analytical procedures.

Authors must include that dispite of the forecasts isolation during the pandemic might have had an effect on the adolescent opportunities of sexual intercourse as described in previous studies, although might have increased sexual violence.

Authors were very discrete in discussing the observed benefits of increasing

PCP coverage in reducing the fertility rate, common to both age groups and this need more consideration, since it is a differencial.

Other issue is the political scenario and program change towards sexual and reproductive health of the country in face of these results. Aren´t they controversial?

Authors would advance if in this section they compare Brazil´s adolesct FR reduction to other countries, since we are following behing. PHC is up since 1994, why the effect is just in the last decade?

R: We would like to thank the reviewer for the comments.

The interaction analysis is described in the methods section, lines 34-36, and presented in the table 3 (previous table 2).

We tried to broaden the discussion, as suggested, although we understand the results of the study only allow us to raise hypotheses.

The effects of the COVID-19 pandemic are probably mixed with less opportunity of sexual intercourse but probable increase in sexual violence. The studies that we cited report an increase in the number of adolescent pregnancies with different reasons for this increase that are already described in the text.

We agree that the results may be controversial due to changes in the political scenario and sexual and reproductive health programs that occurred in the country between 2019-2022. However, we believe that it is beyond the scope of our work to assess the effects of these political changes on the fertility rates. Furthermore, our analysis ends in 2021 and most of the new government (2019-2021) coincides with the pandemic period.

PHC in Brazil began in 1994 and reductions in the fertility rate among girls aged 15-19 and 10-14 have been reported in the country since 2000 and 2010, respectively. Therefore, the effects of increased PHC coverage may have been occuring for a longer time.

Comparing our results with those of a recent paper that assessed fertility rates among girls aged 15 to 19 in 9 Latin American countries with data after 2010, Brazil is not far behind. Our fertility rate among the 15 to 19 age group is the lowest and the rate of fertility decline is the highest, except when compared

---

## [Decision Letter · Decision Letter 1]

Effects of human development, primary care coverage and the COVID-19 pandemic on the fertility rate among adolescents in Brazil in the period 2012-2021

PONE-D-24-43130R1

Dear Dr. Domingues,

We’re pleased to inform you that your manuscript has been judged scientifically suitable for publication and will be formally accepted for publication once it meets all outstanding technical requirements.

In relation to your clarification of software licensing "We would like to inform you that the maps in Figures 2 and 3 were generated by us using the R software, specifically with the geobr and ggplot2 packages. The geobr package provides official spatial data from Brazil, obtained directly from governmental sources such as the IBGE, which are in the public domain and do not have copyright restrictions for use in research and publications. The geobr package itself is licensed under the MIT License. The ggplot2 package, used for visualization and customization of these data, is also licensed under the MIT License. Therefore, the underlying map image is based on public data from the Brazilian government, accessed through a widely used and openly licensed R package, and carries no copyright restrictions for scientific publication. We included this explanation in the methods section and in the footnote of figures 2 and 3.", which was quite thorough in my view, I must inform you that the Editorial Office of Plos One will add to this decision letter requirements as to its understanding of use of the R Package.

Kind regards,

Claudia Garcia Serpa Osorio-de-Castro, Ph.D

Academic Editor

PLOS ONE

Additional Editor Comments (optional):

Reviewers' comments:

Reviewer's Responses to Questions

**Comments to the Author**

1. If the authors have adequately addressed your comments raised in a previous round of review and you feel that this manuscript is now acceptable for publication, you may indicate that here to bypass the “Comments to the Author” section, enter your conflict of interest statement in the “Confidential to Editor” section, and submit your "Accept" recommendation.

Reviewer #1: All comments have been addressed

Reviewer #3: All comments have been addressed

2. Is the manuscript technically sound, and do the data support the conclusions?

Reviewer #1: Yes

Reviewer #3: Yes

3. Has the statistical analysis been performed appropriately and rigorously? 

Reviewer #1: Yes

Reviewer #3: Yes

4. Have the authors made all data underlying the findings in their manuscript fully available?

Reviewer #1: Yes

Reviewer #3: Yes

5. Is the manuscript presented in an intelligible fashion and written in standard English?

Reviewer #1: Yes

Reviewer #3: Yes

6. Review Comments to the Author

Reviewer #1: O artigo já passou por três revisões, as quais atentaram para a maior parte das minhas observações.

Faço um único comentário: a fecundidade adolescente tipicamente diz respeito ao grupo de 15 a 19 anos, a fecundidade do grupo de 10 a 14 anos, apesar de também ser definida como fecundidade adolescente, não pode ser analisada sem uma discussão importante sobre direito. Temos que deixar claro que, na verdade, estamos falando de crianças, em que os dados apontam que, na maioria das vezes, a gravidez nessa idade não é intencional, e frequentemente também está relacionada a situações de abusos e violência sexual. Acho que os autores não podem trabalhar com esse dado da mesma forma que o do grupo de adolescentes de 15 a 19 anos. Como para esse grupo o sistema de saúde faria diferença? Como o sistema de saúde pode proteger essas crianças? E o sistema educacional? Em algum momento, esse tipo de reflexão precisa entrar no texto.

Ainda vejo problema na estima da fecundidade adolescente em nível municipal, não se trata apenas de uma questão de ter zero em um determinado ano, mas da variabilidade do evento ao longo do período. Existem metodologias específicas em estudos demográficos que trabalham melhor esse dado a partir de modelagens estatística.

Reviewer #3: The authors made all the necessary changes and provided clear explanations. The article should be accepted for publication, as it addresses a highly relevant topic and employs an appropriate methodology.

7. PLOS authors have the option to publish the peer review history of their article (what does this mean? ). If published, this will include your full peer review and any attached files.

**Do you want your identity to be public for this peer review?** For information about this choice, including consent withdrawal, please see our Privacy Policy .

Reviewer #1: No

Reviewer #3: No

---

## [Editor Report · Acceptance letter]

PONE-D-24-43130R1

PLOS ONE

Dear Dr. Domingues,

I'm pleased to inform you that your manuscript has been deemed suitable for publication in PLOS ONE. Congratulations! Your manuscript is now being handed over to our production team.

Kind regards,

on behalf of

Dr. Claudia Garcia Serpa Osorio-de-Castro

Academic Editor

PLOS ONE